# Ethical Reasoning in Mathematics: New Directions for Didactics in U.S. Mathematics Education

Jordan Register [1,*], Michelle Stephan [2] and David Pugalee [3]

1 Departments of Mathematics and Statistics and Middle, Secondary, and K-12 Education, University of North Carolina at Charlotte, North Carolina, NC 28223, USA

2 Department of Middle, Secondary, and K-12 Education, University of North Carolina at Charlotte, North Carolina, NC 28223, USA; Michelle.stephan@uncc.edu

3 Center for Science, Technology, Engineering, and Mathematics Education, University of North Carolina at Charlotte, North Carolina, NC 28223, USA; David.pugalee@uncc.edu

* Correspondence: Jtrombly@uncc.edu

**Abstract:** In this article we analyze the current state of the didactics of mathematics in the U.S. as it relates to research on equity. The sociopolitical turn in U.S. mathematics education resulted in a push for critical mathematics pedagogies (CMPs) in which predominantly marginalized populations of students explore social injustices through mathematics. We argue that mathematics education for equity must access broader populations of students and develop students' ethical reasoning skills to build an ethical and equitable future. To address didactical limitations in U.S. mathematics education, we introduce the Ethical Reasoning in Mathematics Framework (ERiM) as a tool to analyze high school students' ethical reasoning in real-world mathematics tasks. Using data from the initial phase of a Design Research project with students of privilege, we show how the ERiM Framework can be used to design lessons that elicit students' critical math consciousness (CMC). We end the article with design implications and recommendations for future research on the didactics of equitable mathematics.

**Keywords:** didactics; mathematics; ethics; equity; U.S. design research; data science

## 1. Introduction

Didactics as a mathematics education enterprise has been alive and well in the United States (U.S.) for decades. Most notably, the National Council of Teacher of Mathematics (NCTM) organization has been supporting both practitioners and researchers since its founding in 1920, while the Journal for Research in Mathematics Education, a premiere research journal in North America, has been an internationally renowned outlet for high quality articles since 1970. According to Lesh et al.'s [1] comprehensive, historical overview of didactics in the U.S., early research focused on the psychological aspects of learning, followed by social aspects, classroom-based research, and teacher preparation in the mid-1980s and 1990s. At the end of their 2014 article, the authors provided some suggestions for future research, but did not anticipate the impact that the emerging sociopolitical unrest in the U.S. would have on mathematics teaching, learning and research. In fact, a short three years later, the NCTM Research Committee published a NCTM Commentary, boldly positioning research on equity, diversity, and inclusion as a professional responsibility [2]. This clarion call to mathematics education researchers, which was first made 12 years earlier [3], involved "eliminating the well-documented inequities of our mathematics education system requir(ing) us to leverage our biggest community asset, our research" [2] (p. 141). In this article, we answer this call by presenting an analytic framework derived from didactic research with high school students as they explained their mathematical reasoning on a variety of problems that provoked some type of ethical dilemma.

The article is organized by introducing a new avenue for research on the didactics of equitable mathematics. First, we analyze the current state of the didactics of mathematics

research in the U.S., especially as it relates to research on equity. Next, we present a brief description of the findings from research on equity, noting significant strengths and areas for further research. We then introduce the Ethical Reasoning in Mathematics Framework that begins to address one of the limitations of current research on the didactics of equitable mathematics. Using data from a Design Research project, we show how the Framework is used and what can be learned about students' ethical reasoning. We end the article with recommendations for future research on the didactics of equitable mathematics.

## 2. Review of Literature

### 2.1. Research on Mathematics Didactics in the U.S.

Based on our reading of didactics articles by authors from a variety of countries, we have seen that the term didactics can refer to either the practice of teaching mathematics or research in mathematics education depending on the cultural and educational background of the writer. For ease of reading, we use the term didactics to mean both teaching and researching and didacticians as practitioners and researchers, clarifying only when needed. Despite different didactics research traditions, Blum et al. [4] argued that European traditions share four key characteristics: (a) didactics includes significant contributions from mathematicians, (b) researchers develop theories for understanding the teaching and learning of mathematics, (c) designers create instruction for mathematics learning environments, and (d) researchers conduct empirical research to show the effect of designs.

Regarding the design of learning environments, Margolinas and Drijvers [5] compared the instructional design approaches of didactical engineering [6] to design research [7]. They found that while each tradition initially had different end goals (practical versus theoretical products), they have undergone a recent shift to include both theory and practice as necessary outcomes. This is also true of mathematics design in the U.S. For example, Stephan et al. [8] provide an account of a classroom-based design research experiment which utilized the instructional design theory Realistic Mathematics Education (RME) to create an innovative measurement sequence for first graders. Drawing on both theory and design to understand their impact on student learning, this study exemplifies how research and practice serve as inseparable companions for practical and theoretical improvements to mathematics education. According to scholars, the most important contribution that Design Research has made is the relevance of research to practice and vice versa. For instance, Gravemeijer and van Eerde [9] maintain that, because design research requires experimentation to be conducted in classrooms as opposed to laboratories, teachers see the use of research to their work. Reciprocally, researchers depend upon the expertise of teachers to build theory on how learning occurs.

Despite the success of Design Research at involving classroom contexts and students' experiences in research, mathematics is still treated as an abstract, non-biased body of knowledge. Sträßer [10], for his part, argued that the didactics of mathematics consists of a didactic triangle of teacher–learner–mathematics and that researchers tend to reduce the human aspect of this relationship by focusing primarily on the institutional setting of school. He contends that students' experiences in the world outside of the classroom are neglected, making mathematics learning appear unrelated to students' lived experiences. Even with the integration of real-world contexts in curricula and research, the relationship that students build with mathematics is that it is non-biased and doable by an elite few. However, in recent years, due in part to the rise of social and political unrest in the U.S., mathematics didactics have reached a turning point.

In a landmark Journal for Research in Mathematics Education (JRME) article, Gutiérrez [11] summarized the 21st century shift that occurred in U.S. didactics research in mathematics as a sociopolitical turn. As acknowledged in the Introduction section, didactics and research began to prioritize perspectives that recognize knowledge, power, and identity as interconnected, derived from, and embedded in social discourses. Didacticians who have taken the sociopolitical turn are concerned with dismantling unjust, taken for granted practices that privilege some and reject others. They not only aim to understand

mathematics education and its underlying social forms, but to transform it in ways that promote socially just practices and outcomes [11]. Evidence for this turn can be seen in the release of numerous equity standards and position statements from mathematics education organizations. The National Council for Teachers of Mathematics (NCTM) released the Principles to Action [12] document which highlighted Access and Equity as one of six guiding principles for mathematics instruction. Similarly, the Association of Mathematics Teacher Educators (AMTE), an organization of professionals who prepare mathematics teachers, released a position statement that establishes its commitment to develop more socially just and equitable systems of education (www.amte.net/positions accessed on 2 April 2021). With a new and promising commitment by U.S. didacticians to equity in mathematics, we now turn to the current state of didactics.

*2.2. Critical Mathematics Didactics (Since the Term Didactics Is Not Commonly Used in the United States, We Simply Use the Words Teaching or Pedagogy in Its Place for the Remainder of the Paper)*

Highlighting the relationship between mathematics and sociopolitical issues is relatively new. A key theoretical perspective that has gained significant traction in the United States is Critical Mathematics Education (CME) [13], which includes pedagogical outgrowths such as Teaching Mathematics for Social Justice (TMSJ) [14–16] and Teaching Mathematics for Spatial Justice (TMSpJ) [17], among others. Such Critical Mathematics Didactics or Pedagogies (U.S. terminology) derive from the Critical Theories of Education that originated at the Frankfurt School in Germany in the 1930s [18]. Frankfurt scholars conducted studies on prejudice and power relations, developing perspectives which are the foundation of critical theories in education today. Critical Theory has been developed by the perspectives of Antonio Gramsci, Jurgen Habermas, Michel Foucault, and especially Paolo Freire, whose ideas about liberatory education guide much of the social justice efforts in U.S. mathematics education [14,16,18]. Given the lack of attention to issues of race and gender in Freire's research, Critical Theorists in the U.S. also draw from areas more inclusive of such topics including Critical Race Theory [19,20], third wave feminism [21–24], and Culturally Relevant [25], Responsive [26], and Sustaining pedagogies [27], among others.

Critical Mathematics Pedagogies (CMPs) are predominantly concerned with social injustices and economic phenomena that affect individual human beings, communities, and society as a whole. According to seminal research by Skovsmose, Gutstein, Frankenstein, and others, CMP pedagogues view mathematical literacy as a necessary component of responsible and critical citizenship and aim to reconceptualize it as critical to the transformation of society [13,14,28]. CMPs have gained significant attention in the U.S. because (1) they engage minoritized students in experiences that they relate to; (2) they may motivate minoritized students to enter STEM fields; and (3) they encourage students to use mathematics to read the world and change it in order to rectify injustices. Given the significant literature which supports the strengths of CMPs both within and outside of U.S. classrooms, we will not elaborate on these points. Instead, we will discuss how the limitations of CMPs and gaps in the literature have created a new avenue for mathematical didactics research which necessarily builds on the following core elements of Critical Mathematics Pedagogies.

CMPs share several core characteristics. First, they dismantle the traditional and hierarchical student-teacher relationship, positioning the teacher and students as co-learners, prioritizing student voice, and effectively emphasizing student empowerment. Second, they argue for a contextualized mathematics curriculum which is based in students' realities (whether at the micro or macro level). For instance, Frankenstein [13] and Gutstein [14] adopt Freire's [29] concept of generative themes, in which students determine the topics of inquiry based on their experiences, communities, and global phenomena. Both considerations create a student-centered approach to learning in which the mathematical content is situated in students' lives. Third, CMPs place an explicit emphasis on the sociopolitical, including power relations between dominant and non-dominant groups both within and outside of schools. This requires that teachers develop what Gutierrez terms, political

knowledge of teaching to support creative insubordination [30]. In her view, this includes a deep understanding of the social and political realities of both the students and the self, enabling the teacher to make creative didactical decisions which support the well-being of students despite systemic constraints in society and education. Fourth, CMPs adopt Freire's position that a liberatory education can only be executed through praxis [14,29]. In Freire's view, praxis is composed of critical reflection and practical action which can only be achieved through discourse [29]. As such, Critical Mathematics classrooms are dialogue and discourse centered. They reject traditional "banking" teaching methods in which teachers "deposit" information into the heads of quiet and submissive students [29]. Rather students and teachers co-create lessons based on their generative themes, decompose, and reflect on the chosen phenomenon using mathematics, and in some cases, use their newfound understanding to become agents of change [14,15]. Finally, CMPs maintain the position that people in oppressed situations cannot be liberated by others but must instead develop the proficiencies required to dismantle systems of oppression themselves. The participants in empirical Critical Mathematics studies typically reflect this viewpoint by engaging homogenous populations of historically marginalized students in a critical mathematical analysis of the systems which oppress their people and communities. While Freire certainly opposed the concept of teachers as oppressors and/pr saviors, he suggested that, to dismantle oppressive systems, the oppressed and the oppressors must work together to transform reality [29].

*2.3. Uncharted Territories in Critical Mathematics Pedagogies*

While teaching with Critical Mathematics Pedagogies and Critical Research has the potential to be highly impactful, from our perspective there are three concerns with current realizations of teaching and researching CMPs: (1) structural and curriculum requirements make it difficult for teachers to commit to a social justice approach, (2) the students engaging in CMPs are typically from homogeneous and/or historically marginalized groups, and (3) CMPs miss the opportunity to explore wider ethical issues, especially those imposed by new mathematics applications such as data science.

2.3.1. Structural Requirements

A major deterrent for the consistent implementation of CMPs is the mismatch between the goals of CMPs and the structural requirements of U.S. schools. Structural requirements that are beyond the control of teachers, such as standardized testing, curricula, and pacing guides, make it difficult for teachers to build their instruction around students' interest [14,31,32]. Both pre- and in-service teachers have reported difficulty balancing mathematical versus social justice goals in their instruction [33,34]. This is not surprising given the general exclusion of social justice goals in teacher preparation. Teacher preparation programs in the U.S. are developed with state and federal curriculum mandates in mind. Despite a developing focus on social justice in some postsecondary institutions across the country, in-service teachers generally have not experienced this.

Complexities related to the implementation of CMPs also relate to the development of social justice curricula and teachers' inability to connect these to their students. Though it is ideal to explore social justice, cultural, and community-based scenarios directly from the students' lives, teachers do not have the time or resources to create lessons that derive from the local context of their school community. In recent years, Critical Mathematics reform materials have been developed for teachers. Consistent with common critiques of CMPs, major findings from Brantlinger's [35] comparative analysis of Critical Mathematics reform curricula were that such texts often replaced mathematical content with political content. This influenced both the rigor of the mathematics to be learned as well as perceptions of who is capable of learning different types of mathematics. While resources such as Gutstein et al.'s *Rethinking Mathematics* [36] and Berry et al.'s *High School Mathematics Lessons to Explore, Understand, and Respond to Social Injustice* [37], act as an effective supplement to the curriculum, it has been noted that teachers must possess political knowledge,

for teaching to facilitate learning of such material [30]. Specifically, teachers' lack of sociopolitical consciousness regarding themselves and their students impairs their ability to facilitate explorations of social justice contexts through mathematics that are meaningful to their unique students [30]. It is imperative that students have access to the contexts that they will explore, especially regarding those concerned with issues of oppression. The consequences of failing to connect these situations to students' lives range from detachment from the learning process [38,39] to reinforcing negative stereotypes [40].

2.3.2. Homogenous Student Populations

A gap in the literature on CMPs as they have been studied so far concerns the homogenous nature of the participants of study, i.e., students with similar racial/ethnic and socioeconomic backgrounds. Of the empirical literature that exists, most of the research has been done with older (middle school and above) historically minoritized populations [14,15,17]. With the exception of studies such as those posed by Esmonde [40] and Kokka [16], one is hard pressed to find studies in critical mathematics education that deal with privileged students (predominantly White students in comfortable socioeconomic positions). This dearth of literature has been explained by Diemer et al. [41] as a consequence of Freire's original development of the critical consciousness (CC) construct. According to their article, CC was created to understand how oppressed people '"read their world" [41] (p. 811). Thus, related scholarship has almost exclusively applied the CC framework to marginalized groups. Diemer et al. posit that the lack of empirical attention to CC amongst advantaged populations may suggest its lack of relevance to privileged people. However, they also suggest that privilege is "relative" in that most individuals have experienced oppression of some sort in their lives (e.g., gender or poverty) and may be able to draw on those experiences to produce change, even for groups who have been oppressed in ways different than themselves.

The concept of relative privilege resonates very strongly with the populations of K-12 classrooms in the U.S. For instance, within each classroom there are a myriad of developing identities and experiences which may, in one context or another, be considered a privilege or disadvantage. Students in K-12 classrooms will have likely experienced relative manifestations of oppression whether as a result of race/ethnicity, socioeconomic status, gender, sexuality, etc. In addition, students are likely to have themselves held (or witnessed) the position of oppressor and oppressed at some point or another. As such, we believe that avoiding working with privileged students misses several opportunities both for the development of CC amongst truly diverse people as well as for research insights.

First, focusing solely on students of color misses the opportunity to encourage students of the dominant race to work on the ethical self [42]. A large part of developing the self is identifying and engaging inn critical reflection on one's own privilege. Mathematics is a useful tool for examining phenomena which deal with the positioning of Whites in relation to other races in society. By encouraging privileged students to examine their position in society and the advantages they hold in comparison to others locally, nationally, and globally, they may recognize the systemic benefits that they not only hold, but also reproduce through their ignorance. Further, promoting critical dialogue and ethical dispositions towards using mathematics solely in homogeneous, underprivileged classrooms fails to explore dialogue that might occur when students are on different sides of heterogeneous classrooms and/or how social justice phenomena can be explored with students in privileged positions. For instance, the findings of Esmonde's [40] analysis of how privileged students (based on race/ethnicity, socioeconomic status, and/or educational privilege) made sense of social justice issues in mathematics contexts suggested that even a mathematical analysis of social justice issues can reinforce harmful stereotypes. The findings indicated that TMSJ curricula brought about significant student disagreements about mathematical solutions which they interpreted through their different personal experiences and backgrounds. Unfortunately, the White, middle-class, academic environment coupled with students' objective mathematical analyses often validated illogical interpretations

(e.g., that people in impoverished neighborhoods are better off), reinforced stereotypes about economically minoritized peoples (e.g., that high poverty neighborhoods require more police), and reinforced privilege (e.g., socioeconomic inequality is a normal part of life) [40].

In contrast to Esmonde [40], Kokka's [16] study of privileged 6th grade students' critical mathematics consciousness yielded more positive results. In her analysis of the rationale that students of privilege use to make decisions based on mathematical data, Kokka found that the students in her study were able to (1) understand sociopolitical conditions, (2) develop critical civic empathy, and (3) take action to change the world (the three components of Kokka's conceptualization of Critical Mathematics Consciousness (CMC) [16] (p. 7). However, she noted that, although these students considered actions that could be taken, "their suggestions were primarily charitable, or they were suggestions for others to take" [16] (p. 18). She therefore suggests that students in privileged positions may need to be explicitly prompted to consider how they will act themselves. As evidenced in Kokka's [16] study, encouraging students to explore their sociopolitical status relative to others in their own community may help privileged students recognize existing structural inequities and develop a sense of empathy. Unfortunately, this may elicit a humanitarian response, or one directed at the actions others should take as opposed to developing a sense of responsibility to empower others. In addition, teaching mathematics for social justice does not always focus on the more general ethical principles involved in reasoning with mathematics. In the next section we discuss current research on ethics in mathematics.

### 2.3.3. Ethical Dimensions of Mathematics Education

A variety of philosophers and researchers have argued for the integration of ethics into mathematics education [42–49]. In an overview of the ethical stances and pedagogies related to mathematics education, Boylan identifies four relational fields which are "mediated through mathematics" [42] (p. 400). These fields, which he refers to as the ethical dimensions of mathematics education, are the relationships with (1) the self, (2) others, (3) the societal and cultural, and (4) the ecological [42] (p. 401).

Boylan highlights a need for an ethical relationship with the self through mathematics education, which he states is intimately related to passion, pleasure, and ethical self-care. He suggests that the mathematics classroom must elicit passionate and enjoyable experiences, intellectual challenges, and critical literacies for the development of "ethical actors in relation to each ethical dimension" [42] (p. 404). In his view, such spaces support the mathematical and ethical development of autonomous actors, who are equipped to fulfil their ethical responsibility to others [42] Caring for oneself is a prerequisite for caring for others. The focus on the relationship with others can be recognized in Atweh et al. [31] and Puka's [43] social response-ability work, which holds that all interpersonal engagement should begin with our ethical responsibility to one another [31] (p. 401). Based on Puka's [44] feminist writings, the distinction between responsibility and response-ability is an ethical one, where response-ability refers to a person's ability to "respond to the demands of our own well-being" and to the "demands of the other" [31] (p. 269). In a similar vein, the societal and cultural dimension is related to the view that mathematics is a social and cultural practice that is a "product of our ancestors" [42] (p. 402). Critical Mathematics education scholarship relevant to the sociopolitical turn [11] in mathematics education begins to deal with this dimension through its discussion of social justice, critical mathematical citizenship for participation in democracy, and agency for addressing both [13,14,16,28]. However, Boylan suggests that there is a dearth of ethical discourse in this area [42]. Finally, the ecological dimension considers issues connected to globalization and human survival [42,45]. It includes the consideration of environmental factors and an ongoing critique of mathematics as a social construction, separate from nature [42]. According to Boylan, ethical reasoning concerns issues of fairness and our personal choices but also must consider relationships beyond those which we are directly connected to [42].

Given the ethical nature of the environmental crisis, data science, engineering, and the financial disciplines, among others, it seems irresponsible to exclude ethical reasoning and design (i.e., how products/solutions may affect individuals, communities, society, or the environment) from 21st century mathematics curricula. Current and future research should explore the institutional constraints on teaching with CMPs and conduct design research to create mathematics programs that integrate CMPs with the standards that must be taught at each grade level. Furthermore, design research should explore CMPs in heterogenous contexts as well as determine how to develop ethical reasoning in mathematics in both heterogeneous and homogenous populations. We turn now to a framework we are developing to perform such research, the Ethical Reasoning in Mathematics (ERiM) Framework.

### 2.4. Ethics and Social Justice

In a 2014 Editorial, Stinson [47] questioned the failure of the mathematics community to permanently adopt social justice mathematics into the "canon" of mathematics teacher education and as an integral part of the Standards for Mathematical Practice. With students' increasing awareness of past and present social injustices (through technology and social media), he questioned why an ethical and moral responsibility in engaging students in authentic problem solving of such issues is not assumed? He suggested that choosing not to engage students in social justice mathematics implies a failure "to uphold our ethical, moral, civic, and pedagogical responsibilities" [47] (p. 4).

It has been further argued that "ethics in mathematics education supports, and lays the foundation for, concerns about social justice" [31] (p. 268). As such, scholars concerned with ethical mathematics education maintain that ethical mathematics is critical mathematics [47–49]. Atweh et al. explain that issues of social justice are typically concerned with the social activity of groups of people and the fair enjoyment of social benefits while issues of ethics are concerned with interactions between people [31]. From this perspective, ethical considerations are based upon people's moral responsibility to one another, establishing "social justice concerns as a moral obligation, rather than charity, good will or convenient politics" [31] (p. 268). While "knowledge serves justice", "justice serves moral imperatives," positioning ethics and social justice as separate but interdependent realms [31] (p. 268).

According to Atweh et al., those who subscribe to the transformative role of mathematics education, see it as a means to "create the world in a new way" [31] (p. 270). This goes beyond responding to injustice (analogous to most of the work done surrounding mathematics for social justice) and instead focuses on responsible creation of mathematical products. Borrowing from Freire's concept of a liberatory education, Atweh et al. argue that mathematics education should serve to develop students' ability to both "read" (i.e., understand) and "write" (i.e., change) the world [31]. They propose that to prepare students to transform the world requires that they develop ethical and social response-ability. In applying this concept to mathematics education, Atweh et al. noted that the primary aim should be to "enable the response-ability of students in their current and future lives as citizens" [31] (p. 270). As such, a response-able mathematics education must go beyond the exploration of current or past injustices, but should prioritize the development of students' ethical reasoning, needed prior to creating the world, with mathematics. In the sections that follow, we humbly outline a new framework, grounded in Critical Theory and Ethics, which may be used to understand the ethical reasoning of mathematics students.

## 3. Materials and Methods

### 3.1. Ethical Reasoning in Mathematics (ERiM)

The ERiM framework is situated within the Critical Mathematics Education program elaborated by Frankenstein [13] and Shor [50]. The complementary goals of CME are to promote students' critical mathematics reflection so that they can question mathematical arguments, models and representations, and inspire critical mathematics agency to act in ways that use mathematical communication to liberate, rather than disenfranchise, groups.

Kokka [16] defines Critical Mathematics Consciousness (CMC) as the development of sociopolitical understanding, critical civic empathy, and action-taking through mathematics. Our definition of CMC differs in that it emphasizes mathematics through a critical lens. Critical mathematics consciousness, for us, refers to the awareness of the role that mathematics plays in disenfranchising or liberating oppressed groups in society and the willingness and commitment to act (i.e., critical mathematics agency). CMC involves three different types of mathematical awareness (MA) (authors, accepted for publication):

- The Ethical Mathematics Awareness that human beings do mathematics; thus, there are potential ethical dilemmas and implications in mathematical work.
- The Communicative Mathematics Awareness that mathematical communication has the power to educate and mis-educate society and encourage the masses to act in certain ways.
- The Sociopolitical Mathematics Awareness that mathematics is used to model and interpret the real world and can be used to make decisions both at the individual and systemic levels that may be oppressive or liberatory.

Figure 1 illustrates the Critical Mathematics Consciousness theoretical framework that guides our research. Each of the three types of awareness are included along with the name of an analytic framework used for data analysis. We have presented the Sociopolitical Analytic Framework in another article [51] and have yet to formulate a framework for Communicative Mathematics Awareness. In this article, we present the Analytic Framework that is used to understand students' ethical mathematics awareness.

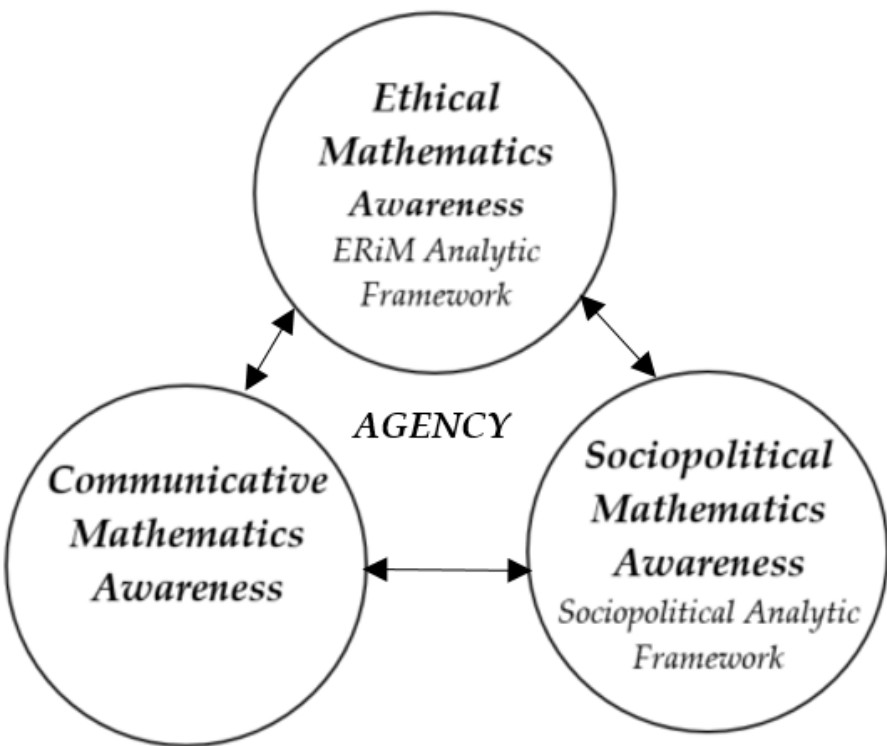

**Figure 1.** Critical Mathematics Consciousness Theoretical Framework.

### 3.2. Ethical Reasoning in the Mathematics Analytic Framework

Ethical mathematics awareness refers to being aware of ethical implications that may or may not be considered in the process of making mathematical decisions, such as violations of privacy, questions of who owns the mathematical data, and bias in the data, to name a few. To create the Ethical Reasoning in Mathematics Framework (ERiM) in Table 1, we consulted a variety of resources including philosophy and mathematics education articles. While the general principles expressed in these resources were useful, we also

suspected that more nuanced principles pertaining to the STEM professions might be found in materials published by professional organizations for engineering, data analytics, artificial intelligence, computing and information science, and statistics. Since many of our interview questions employed a business scenario as the context for making decisions, we also explored resources from business domains including marketing, accounting, and finance. As we read each article, ethics statements from organizations and philosophical writings on ethics in mathematics, we noted the particular principles that were named across professions. When considering business situations and more recent STEM disciplines such as AI, Computing, and Data Sciences, additional principles began to emerge involving the ethics of how algorithms and data analysis results might be used to impact society. In Table 1 below, the left column lists the ethical principle that was named in the documents we read, the middle column are examples of questions one might ask as we consider that particular principle, and the right column indicates which document and/or professional discipline states the principle. As an example of how to read the table, consider that privacy issues are of main concern in mathematics in multiple ways. In obtaining data for a business client or a biology laboratory, has consent been given by the people whose data has been collected? Have people been coerced into giving that data or was it given freely? How have the data collectors ensured privacy is not violated when they present their findings to the public? Most of the STEM disciplines we explored named privacy as a major ethical obligation. As can be seen in the Table, privacy, fairness, accuracy of data, and accountability are the most prevalent ethical obligations considered by STEM and Business professions.

**Table 1.** Ethical Reasoning in Mathematics Analytic Framework.

| Ethical Principles | Considerations | Professional Domains Addressing the Principle |
|---|---|---|
| Privacy | How does respect for freedom or personal autonomy apply? Is confidentiality required? Is consent needed or obtained? Is privacy violated? | NSPE [1], JMU, ASA, ACM/AI |
| Fairness (equality) | What is the fair or just thing to do? Is there fair access to systems that are created? | ACM, AIB, Data Analytics, JMU |
| Accuracy | Is data reliable and accurate? | NSPE, AIB, ACM/AI, Science, Data Sciences |
| Accountability | Who is accountable? Have they communicated the data in a misleading manner? Is the source reliable?) | NSPE, AIB, ACM/AI, Science, Data Sciences |
| Property | Whose data is it to sell? Who owns the data? | Data Sciences |
| Loyalty | Is the decision/activity loyal to the organization (e.g., makes profit, keeps ideas within organization, does not use organization's ideas to make money elsewhere) | AIB, JMU |
| Accessibility | What information does an organization have the right to access about people? Who has access to this data? User? Buyer? | Data Sciences |
| Algorithm Bias | Are algorithms objective? Do algorithms (un)knowingly discriminate against individuals or groups? | AI |
| Transparency | Are the codes for algorithms readily available for inspection? | ACM/AI, Science, ASA |
| Ecological | Has the impact on humans and ecosystems been considered? | ACM/AI, NSPE |
| Employment | Will the decision/activity harm an individual's or group's employment status? | AI |
| Discrimination | Has the decision/activity avoided negative effects on oppressed societal groups? | Data Sciences, ACM/AI, AIB |

[1] James Madison University Ethical Reasoning in Action (JMU); American of Statistics Association (ASA); Science [52]; Data Sciences; National Society for Professional Engineers (NSPE) [53]; Association for Computing Machinery (ACM); Artificial Intelligence (AI); International Business (AIB).

### 3.3. Participants

The research presented in this article was conducted as the first step in a Design Research Project [7] that focuses on Designing Critical Mathematics Consciousness (CMC) for Students' of Privilege. In the initial phase of a Classroom Design Research Project, interviews can be conducted when there is little research on students' understanding of the topic. Thus, interviews were conducted with 14–16-year-old students with economic and racial privilege to answer two related research questions: (a) what is high school students' critical mathematics consciousness and (b) what types of contextual problems may provoke aspects of Critical Mathematics Consciousness? The three schools were chosen as a matter of convenience. The researchers had personal relationships with the administrators and obtained permission to interview very easily. Study participants learned of the interview opportunity through presentations by a researcher in their mathematics class and volunteered. The Design Research Team consisted of two mathematics educators, two STEM educators, one doctoral student and one assessment professional. We interviewed thirteen students, nine of whom identified as male and four as female. Nine of the students matriculated at Hill High Charter high school, three from a Lakeview Charter middle school and one from Paradise Bluffs, a private middle school. Thirty percent of Hill High Charter's students perform at grade level in mathematics and the student population is 65% White, 22% Black and 5% Hispanic. Sixty-three percent of Lakeview Charter's students perform at grade level in mathematics and the student population is 70% White, 13% Black, and 5% Hispanic. Finally, eighty percent of Paradise Bluffs' students perform at grade level in mathematics and the student population is 75% White, 13% Black, and 1% Hispanic. All data collection procedures were approved on February 17, 2020 by the University of North Carolina (UNC) Charlotte Institutional Review Board (IRB) (IRB Number: 19-0594) and all permissions were granted in writing by participants and their parents/guardians prior to interviews.

### 3.4. Data Collection

The interview tasks were designed to elicit students' Critical Mathematics Consciousness and agency. Interviews lasted from 25–45 min and took place at the student's school in a private setting during a non-academic class period. One member of the research team conducted the interview while a second team member took field notes and video recorded the session. The interviewer began each session with general talk, attempting to build rapport with the student. Then, all students but two were asked to reveal their thinking on each of five tasks (https://bit.ly/mathethics accessed on 2 April 2021). Although there were five total tasks, *Corona Crisis* and *Great Groceries* were the most relevant for this analysis because they elicit students' general ethical awareness.

## 4. Results

To analyze the data, we first transcribed all 13 of 15 interviews. Since the two interviewers were White and unfamiliar to the students, we were concerned that Black students may have been uncomfortable sharing their genuine feelings about race. Thus, the data from the two Black students from Hill High Charter were not used in the analysis. The remaining Black students were friends of one of the interviewers, and we were confident that their interview responses were reliable. We used the general CMC Theoretical Framework to guide the analysis. First, we used the Ethical Reasoning in Mathematics Framework to classify any ethical considerations made by students in their decision-making process. Second, we documented the CMC awareness types that arose as they analyzed each problem situation. Three Design Team members independently coded the interview transcripts and met to determine consensus on codes that differed across researchers. In the next sections, we present our findings for the *Corona Crisis* and *Great Groceries* tasks.

*4.1. Corona Crisis*

For the *Corona Crisis* task (see Figure 2), we capitalized on the current COVID-19 pandemic that had just become major news in the United States when we began interviewing students. In fact, three days after we finished the last interview, the state issued a Stay-at-Home mandate as incidents of infections began to rise significantly. *Corona Crisis* is presented in two parts. In the first part of the task, the student was presented with the image of a tweet from the president of a fictitious bank, who was using the data to suggest that infections were declining. The student was asked if she would take his advice and whether (and how) she used the graph to make that decision. In the second part of the task, a second graph was shown with an image of a post on the fictitious Center for Health and Disease Control's (CHDC) Instagram account, with the statement that "the disease is still a high risk and there is a Level 1 warning." Students were told that the CHDC used the same data set from the fictional Darden Data, Inc., and were asked to compare posts to determine whose advice they would follow and why.

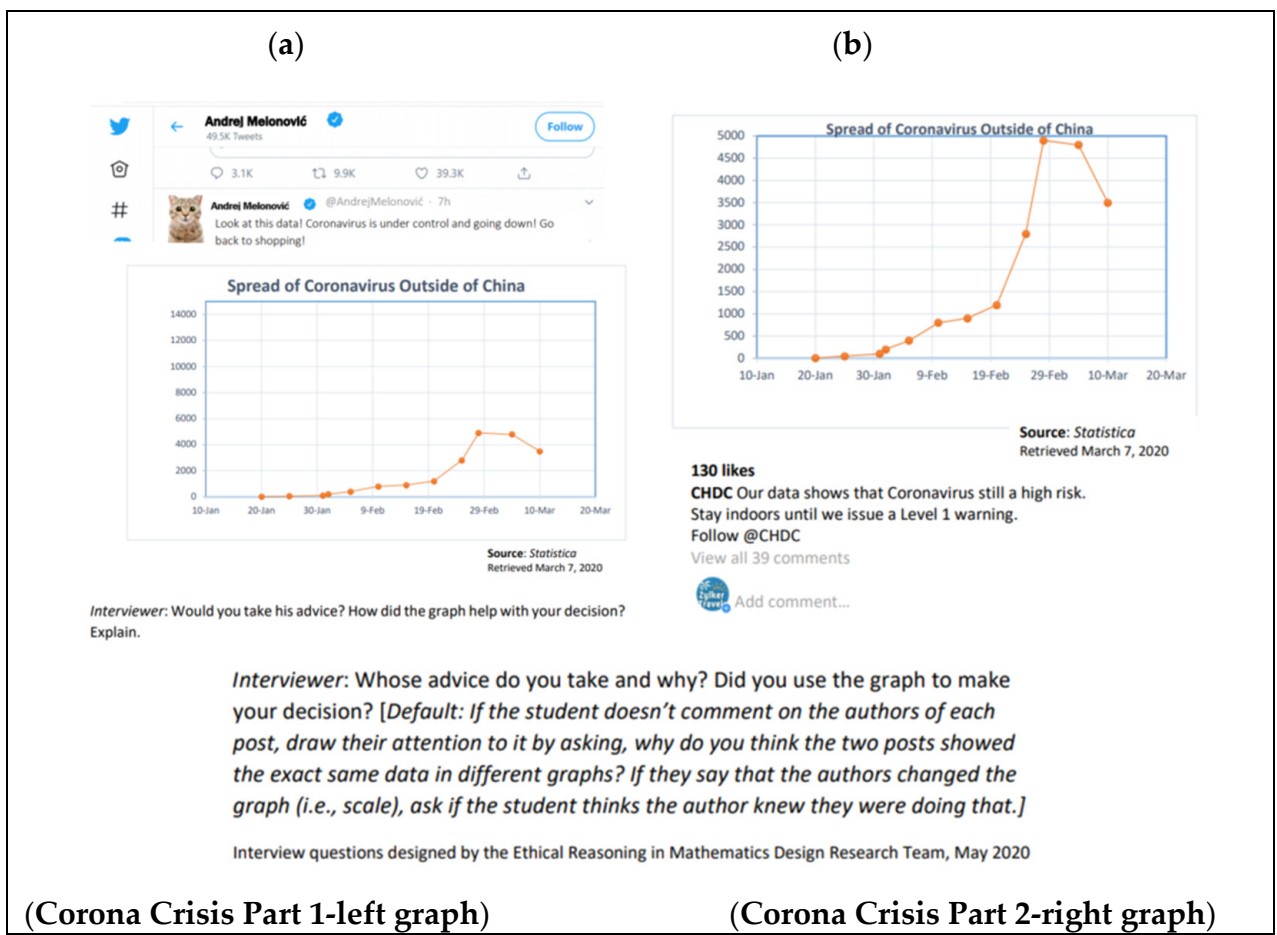

**Figure 2.** (**a**) Corona Crisis Part 1 depicts a Tweet from the president of a fictitious bank, using the data to suggest that infections were declining; (**b**) Corona Crisis Part 2 depicts a post from the fictitious Center for Health and Disease Control (CHDC) using the same dataset to claim that the disease presents a high risk.

*Corona Crisis* was designed to elicit students' mathematical interpretation of graphs as well as their awareness of the communication strategies used by people who create graphs. Specifically, it was designed to explore students' awareness that mathematics can be used to manipulate society into thinking and behaving in ways that may benefit or disadvantage others in society(communicative MA) as well as how students' mathematical understanding of scales affects their interpretations and subsequent decisions. Since we had not designed the *Corona Crisis* task prior to starting the interviews, there are data on

only 11 students. Three main themes emerged from analyzing students' responses to the task: (1) reasoning with dynamic graphs, (2) questioning accuracy and accountability in data dissemination, and (3) exhibiting communicative mathematical awareness.

### 4.1.1. Reasoning with Dynamic Graphs

When critiquing the first graph, all but one student used language that indicated that the graphs were still changing; that they were living, dynamic pictures. Consider several reactions by students when asked whose advice they would take. All italics are added for easy reference.

*Collin*: I feel like not yet (going out shopping). Once it *gets down* to like almost to zero and you can make sure everything's okay. Just to be 100% sure that it's not around or any cases are there.

*Matthew*: But I mean it does look kind of under control, but you don't know *what could come*. You have like March 30. And then you have April 10 and then that could be like (*swipes a finger upward from the end of the graph as if it would skyrocket again*) right back up.

*Tony*: I think he's (banker) speaking a little bit too soon. Cause just because the Coronavirus stats *dropped* one time doesn't mean it's not gonna *jump back up*.

Students' specific wording above indicates that they view the graphs as changing, likely because the pandemic had just begun, and the students knew that the data was being updated almost daily. Research has shown that students generally view graphs as a static correlation between quantities [54–57] so this task provides an opportunity to confront such notions. The Coronavirus context also has the unique possibility to elicit students' thoughts about the predictive power of mathematics. The predictive power of models, in fact, is the very reason that attention to mathematical models has appeared so prominently in the public discourse around COVID-19. The model is used to describe the infection rate pattern that currently exists so that discussions can center on ways to "flatten the curve." Such language in both the public and students' discourse indicates that this type of model is alive and can change. That is, if the conditions under which the number of infections is decreasing continue, the data should remain on a downward trend unless something affects the infection rate. It is not clear if the students thought that the trend of the curve was simply the natural progression of the virus or if it could be affected by social action such as the "stay at home order." However, the students did recognize that mathematics can be used to model the real world and that they can interpret it to make an informed decision.

### 4.1.2. Questioning Accuracy and Accountability in Data Dissemination

Students demonstrated concern for the accuracy of data and the accountability of those who report it. Close to half of the students interviewed examined the source, reliability, and/or relevance of the data represented in each graph. Several inquiries of this nature were concerned with the accessibility that the authors of the graphs had to the data, and their authority to make claims. For instance, Charles and Rene suggested that the *Corona Crisis* graph was not representative of the entire dataset. While Charles doubted the accuracy of the data, Rene recognized that the crisis is ongoing and that the graph could change.

*Charles*: Because, how does he know, because the graph can't tell you the truth, I guess. It doesn't, there could be cases somewhere in the world that we don't know.

*Rene*: So, like the data is still changing. You can't just take information from one piece of, like evidence. So, like, I mean you could, but like it might not be 100% accurate.

### 4.1.3. Exhibiting Communicative Mathematical Awareness

The final theme that emerged from students' explanations concerns both their mathematical reasoning and their awareness of how individuals manipulate graphs to persuade society. Significantly, all but one student noticed that the banker and CHDC graphs contained the same data but used altered visualizations. For a few students, the observation

was entirely visual-spatial where they described the difference as one of zooming in or out, much like the zoom function of a graphing calculator or computer. All students, however, reasoned mathematically by attending to the increment size on each scale. Furthermore, the student that did not notice the data was the same took the advice of the CHDC. In fact, the source of the graph was the primary basis upon which a majority of students made their decision about staying at home. However, when asked if they thought the person (source) who created the graph was aware that he was changing the way the data was presented, all of the students but one said yes.

*Jade*: I feel like he's (banker) definitely saying that because he wants to make profit and because we . . . we watched like the CNN 10 thing (television news station for students) and like every day in my social studies class which is really interesting...but they had a whole one (video segment) about how it's affecting the Chinese economy specifically and how it could start to affect our economy which I thought was really interesting. So, I don't blame him for saying that but that isn't a very honest thing...and he doesn't have the authority to say that.

*Tony*: He might have made his graph on intervals of 2000 just make it look like it's going down now. He might have tried to trick some people knowing that the internet is not a very smart place...I mean, he's showing real stats. I think that he's self-interested. He wants you to go shopping. He doesn't really care about your health . . . (Interviewer: What about CHDC?) I guess they're just trying to protect you. I would trust them.

Both Tony and Jade recognized that the scale increments had been adjusted and that this technique was being used blatantly by the banker in order to trick the public to go shopping or to profit off people. Jade went so far as to question the banker's honesty but understood the pressure he was under to manage the economy.

While most students relied on the CHDC, two students argued that the CHDC was attempting to scare the public into staying at home by using inflammatory words like "Level 1 warning" and "high risk" whereas the banker was trying to make you feel safe.

*Nicholas*: Well, I wouldn't go out immediately. I'd still wait like for news sources to like die down and stop overreacting. This one (CHDC) is made to look more scary. Well (it has) just a bigger spike, but the same numbers if you compare the two charts. But yeah, I would be more scared of this one (CHDC). No, this one (banker) makes it look less dramatic. But I do think this one (CHDC) is intended to strike more fear into people. I'd probably go with this one (banker) if I had to like go out and shop just because this (CHDC) looks like a more dramatic fall.

*Charles*: I scroll through here (mimicking scrolling on the phone image) and I see that: There's *warning*. The word *warning* kind of sets it off. It's scary....And so that makes you kind of want to stay indoors. (The CHDC graph was modified) to let this stress off everyone.

These excerpts show how some students recognized that the graphs and, as importantly, the words used in their posts can evoke emotions one way or the other. Therefore, a majority of the students recognized that mathematics, mathematical representations, and the language that accompanies them have the power to influence society's beliefs and/or actions.

In summary, all students were able to access the mathematics in this context easily and used it to "read their world." On the other hand, most students indicated that they did not trust the data or the sources (accuracy and accountability) to predict the behavior of the virus, which calls into question the students' belief that mathematics can be used to model the real world accurately. Additionally, all but one student recognized that people could present the same mathematical data differently to manipulate society's actions, thus exhibiting strong communicative MA. Although this question was asked as the last question of the interview, we realized that it might have been better positioned as an opening task. While we will elaborate on this point more in the Discussion section, it is important to note here that *Corona Crisis* was accessible to all students (mathematically and contextually), it evoked important ethical questions related to the value of profit in

our culture, and it elicited students' thinking about the relevance of mathematics to the decision-making processes that affect human beings and society as a whole. These themes extended into the *Great Groceries* Analysis that follows.

*4.2. Great Groceries*

The *Great Groceries* task (see Figure 3) was developed to explore which ethical considerations students demonstrate when making economic decisions based upon the results of big data analyses, a topic of concern for data scientists. The response options were designed intentionally to create a potential ethical dilemma for students between choosing an option that placed profits over people. Option A may prompt students to recognize issues of economic classism and structural oppression by imposing a higher shopping fee on the poorest customers. Option B, in our view, posed the least harm to people but also raised the least profit. Option C would generate the most profit but also eliminate jobs. When a student made a decision, the interviewer posed follow-up questions to elicit justifications. Each interview was transcribed, and three team members used the Ethical Mathematics Reasoning Framework to categorize each student's responses independently. Once coding was completed, the three team members met to compare results and achieve consensus.

Great Groceries is one of the biggest successes in American grocery chains. Impressively, Great Groceries reported $14.74 million in quarterly profits last year. You are the CEO for Great Groceries. One of your goals for 2020 is to increase profits. Your Management Team comes up with the following three recommendations for increasing profits. Which option would you recommend to your CEO, if any? Explain

**Option A:** Offer tiered membership fees, the idea being that richer customers buy more products. The Tiered-Membership Program would increase profits by 1.8%.

|  | **Member's Yearly Income Level** | **Membership Fee per Year** |
|---|---|---|
| **Upper Tier** | $150,000 and higher | free |
| **Middle Tier** | $70,000-$150,000 | $50 |
| **Lower Tier** | $30,000-$70,000 | $75 |
| **Sub Tier** | $Under $30,000 | $100 |

**Option B:** Raise the price on the five top selling items by $1.50 each, increasing profits by 0.5%

**Option C:** Put more digital, self-serve checkout stands in their stores to decrease the number of human workers, raising profits by 3.5%

**Figure 3.** The *Great Groceries* interview task.

Thirteen of the interviewed students answered the *Great Groceries* problem. Of those students, one Black student chose the tiered membership option (A), six students (three Black and thee White) chose the top sellers option (B) and six White students chose the self-serve checkout option (C). The participants' rationale was guided by either profit or their morals/ethics, split evenly amongst the group. In the following sections, we will discuss the students' reasoning according to these major themes as well as the sub-themes that emerged within them. For profit, the sub-themes included loyalty to business and making profits and loyalty to the employer. For decisions made based on ethics, the sub-themes included fairness, discrimination, employment, and loyalty to workers and the economy. We will conclude the analysis by discussing how students used the Coronavirus pandemic to make sense of Option B in the scenario.

4.2.1. Profit

Half of the students made decisions based on a desire to increase profits. All six students who ultimately chose Option C, to install self-serve checkouts, made their final decision based on the fact that it would raise profits the most. Interestingly, the desire to maximize profits was made under the guise that profit is the ultimate goal of the business world, despite any collateral damage. For instance, in discussing his decision-making process, Tony cited profit as the overarching value when running a business.

*Loyalty to Business and Making Profits.*

**Tony**: I would just do Option C. Because the increase in profits is more than any other one. I think when you're running a business, that's what it comes down to is profits.

Additionally, while he recognizes that Option A is unfair, he connects this back to his economic goals, noting that it will be controversial among customers, and thus bad for business.

**Tony**: I feel like Option A would be controversial between the customers because they are being unfairly treated. One of them is having to pay nothing to get into the store and one of them has to pay $100. I feel like it's not gonna be good for the business. From a business standpoint, it works. But I don't think any of your sub-tier people are gonna pay $100 if they know that people are just getting in for free because they make more a year.

**Interviewer**: What about Option B?

**Tony:** It only raises your profits by 0.5%. I just think that 3.5% is a better profit raise than 0.5%.

Ultimately, Tony chooses the option that will make the company the most money. In a similar vein, Collin chose Option C to increase profits, noting that by installing artificial intelligence, he will not have to pay the workers that it will replace. When he voiced some resistance to the idea that people would lose their jobs, the interviewer asked if he would respond differently if he reported to himself as opposed to the CEO (several students missed that they held the position of CEO in this scenario). Given the option, Cooper chose profit over job security for his workers.

**Collin**: I guess C. Because, it's kind of hard because you're also putting so many people on the jobs. But that would also mean you wouldn't have to pay them. You wouldn't have to pay them at all. So yes, your profit would go up. So I guess, C.

**Interviewer**: So what if you didn't have to report to the CEO, but you were able to make that decision? Would you, would any of your decisions change? Would you still choose options C?

**Collin**: Option C.

*Loyalty to Employer*

In contrast to Collin's choice to stick with Option C, two students changed their minds if they did not have to report to the CEO, an interesting finding which we will elaborate on in the Synthesis of Analyses section. While Ella originally chose Option C to maximize profits, she eventually went with Option A because digital, self-serve checkouts leave out the human element of shopping and might decrease profits. According to Ella, "people like to talk to humans when they are at the store," which she thought might be a problem for the company.

Jade, on the other hand, switched her decision from Option C to Option A, citing her moral discomfort with people losing their jobs.

**Jade**: I'd probably say the first one (Option A) then because, um, well you're not forcing everyone to pay an extra dollar 50, and if you can afford the membership, then you'll get the membership. And then you're not worried about people losing their jobs or not being able to get jobs.

Jade's rationale for choosing Option A over her original choice, Option C, mimics the reasoning of the second group of students, who made their decision based on what many of them referred to as their morals. Several of the students explained that there were two different avenues that they could follow in this scenario: the moral way or the way that

made the most money. One student, Tyrone, even discussed the implicit nature of business decisions being inherently immoral in certain situations.

*Tyrone*: Um, out of all of these options, B makes the most sense. And I actually kind of like that idea cause it's either way being in a business and increasing your process or your profits, you're kind of cheating out and being selfish any way you go. That's just basically being a businessman. You're losing your morals anyway. But because all of these are really like disrespectful, but you want more money.

While students like Tony and Collin ignored their moral reservations and chose the "best" business decisions, Tyrone and others could not. In the following section, we will discuss these students.

4.2.2. Ethical Reasoning

For the participants who chose Option B, their decision was based ultimately upon their personal ethics. However, they demonstrated a diversity of ethical considerations in their decision-making process. This included fairness, discrimination, employment, loyalty to workers and the economy.

*Fairness*

All six students said that Option A was not fair since some members are required to pay more than others. Consider Mai's response to Option A, where members of different income levels would be required to pay different amounts.

*Miya*: If I were, let's say sub tier here, and I had low income . . . . I want a membership. It's $100. And then . . . I think I see this other person who's an upper tier class and I ask them how much they pay and like, Oh yeah, it's free. Like I don't know. I would not go to a great grocery then. It's not great. Just be bad. Bad Groceries! (laughter) Yeah, why should it be on tiers? No, it should be equal.

In this scenario, Miya equates fairness with equal membership fees despite income. She expresses her dissatisfaction with having to pay something different than someone else. Tyrone and Jamal, the two African American males in the group, integrated the ethical element of fairness with issues of employment and discrimination. Both students identified that wealthy people are being asked to pay less than those with a lower income (employment). In addition, Tyrone noted the relative expense of the membership fee to those in the lower tier, recognizing that it could put them in a difficult financial position (discrimination).

*Jamal*: I pick Option B cause like, um, Option C, taking people's jobs didn't sound right and because people need money too, to help their families. And Option A, just helping the richer customers is, it's not really cool...Well that's not really fair. Like the people with more money get free stuff.

*Tyrone*: Option A, definitely not...you're cheating out the lower class... they have to pay a membership fee of $100...that's a lot of money when you have $30,000 to spend every single year...Option B, You're not going to get as much money as you want compared to the other two options. But that's the most fair route I would go because everything else is just like morally incorrect... You can't do that. That's mean... because you're taxing, um, poverty.

*Discrimination/Sociopolitical Mathematical Awareness*

Tyrone further evoked the ethics of discrimination in deciding not to choose Option C, to install self-serve checkouts. Specifically, he recognized that a consequence of artificial intelligence in this scenario is the loss of jobs for unskilled workers.

*Tyrone*: Option C, that decreases job options for people. And I don't want to decrease jobs because people still need jobs. And, um, I know a lot of the grocery store occupation is like . . . people in poverty, high-schoolers and retirement, people in retirement. So I don't think it's fair to any of those groups because they're the ones most troubled, to take away the jobs cause that it may seem like only one store. But if you're going to do that worldwide to your whole company, then that's bad.

Tyrone's consideration of the types of people who compose the unskilled workforce and the effects of mass unemployment for such people, speaks to his developing sociopolitical awareness (the first characteristic of CMC) in regard to the economy.

*Employment*

The ethics of employment resonated with each of the other participants who rejected Option C. All of these six students cited reasons concerned with job loss, albeit in slightly different ways. Though she did not discuss it on a systemic level like Tyrone, Greta expressed her concern for limiting jobs for unskilled workers.

*Greta*: I think I would go with B because option C you would kind of be putting people out of jobs when people, like might need them, and...a cashier is like, kind of a simple job, but it's not. But it pays you.

*Loyalty to Workers/Building the Economy*

Using the same rationale as Greta for dismissing Option C, Charles chose Option B out of his perceived loyalty to the workers. In his view, increasing profits for the company would translate to higher pay for the employees of the company.

*Charles*: I think Option B. Increasing profits means more money for the workers, I guess. And (not Option C) because it gets people out of their jobs.... (It's bad because) they're not working. It (Option B) increases profits. Just raise the price of the top most popular item sold. So yeah, it makes sense.

In contrast, Miya dismissed Option C because it limits the number of workers who can stimulate the economy.

*Miya*: So, we need workers. I think it's important for our economy because it helps cause... the more people who work... are helping the economy . . . the less people that work...they wouldn't build anything. They would just stay there. It's like they're sleeping. So neither are they building it or doing anything to it. So if less people, less and less . . . did that work, I feel like our economy would plummet. Our economy right now is already horrific. Like we're in 3 trillion debt right now. Yeah. It's awful. So the more workers... more money spreads out . . . there's taxes too...and like the people who are paying the workers . . . they go out (and) buy stuff, it's like a whole cycle. And so the more people who are in this cycle, I feel, the more money the government can get and they can pay off this and then the economy would burst and more ideas would spread and just like, could start industrializing again. Maybe.

Interestingly, Miya continued her decision-making process by drawing from the impact that the Coronavirus pandemic was having on the U.S. at that time.

4.2.3. Connecting to the Coronavirus Pandemic

In contemplating what she felt was the best option in the scenario, Miya drew on her understanding of the toilet paper crisis that was beginning to occur in the United States.

*Miya*: I would say B, it seemed like it's not that bad, depending on what the five top selling items are. Cause . . . let's go back to toilet paper. Very important... right now it's one of the top most sold objects, all because we have the coronavirus . . . We (*Great Groceries*) have a good amount of toilet paper, but if just one person's going to buy all of it and then proceed to, like, sell it for double the price, cause people are doing that on like Craig's List (an internet sales site) and all that, which is crazy.

Similarly, Greta used the pandemic to help her understand the meaning of Option B (increasing the price on the top five most sold items) in the scenario by connecting it to the price gouging of hand sanitizer.

*Greta*: (Option B) makes a lot of sense cause I see that a lot... We're actually seeing it now a little bit like hand sanitizers... Because of the virus... everybody's raising the prices on it cause everybody wants more of it. And if it's five top selling items, then it's most frequently bought. So more people will buy it and then it goes up in price.

While their connections to the pandemic did not necessarily influence their ultimate decision, we feel that Miya and Greta used it to mediate their understanding of the context by connecting it to their experiences.

### 4.2.4. Synthesis of Analyses

The analysis of these tasks presented several interesting findings related to students' Mathematics Consciousness (CMC) and their ethical reasoning. By developing tasks which promote and draw on students' sociopolitical, ethical, and communicative awareness (CMC), these two tasks alone elicited students' consideration of six ethical elements in the ERiM Analytic Framework. Specifically, *Corona Crisis* encouraged students to consider issues of accuracy and accountability in data dissemination, while the *Great Groceries* task drew out students' concerns with fairness, discrimination, employment, and loyalty.

The Corona Crisis task elicited students' consideration of the motivational difference between business owners and health officials and the effects of those motives on the behavior of the general public. Through these discussions, students were able to elaborate on how mathematical representations can be used to persuade citizens and affect their perception of phenomena as well as promote the interest of organizations, corporations, businesses, governments, and other bureaucratic institutions. Specifically, they referenced how titles, descriptions, and the scales used on graphical representations influence how consumers perceive a given message. While the Corona Crisis task drew on and developed students' communicative mathematical awareness, it did not require any agency per se. Rather, it encouraged students' recognition of the subjective properties of mathematics. In contrast, the Great Groceries task required that students apply ethical principles in using mathematics, as elucidated in the ERiM.

Our concept of an ethical mathematics education includes that students develop ethical response-ability. *Great Groceries* is an example of a task that encourages students to make decisions using results from big data analyses that have ethical consequences. In general, students made their decisions based on profit or ethics, albeit in very different ways. An interesting finding in our analysis of Great Groceries was the distribution of loyalties to workers, the economy, and business authorities. For instance, some students made their decisions according to what they perceived as good business practices, while others made their decisions based on their "morals." Such morals, which we consider to be their personal ethics, allude to their concern for fairness, discrimination, employment, and loyalty in the business world. In addition, several students expressed that they would make different decisions depending on who they were reporting to: if they reported to a CEO they would make the better business decision (i.e., most profit), whereas if they were to make the decision for themselves, they would choose the moral path. It is also interesting to note that some students did not feel that they had a choice when reporting to an external CEO. This poses important questions about how students perceive the ethical responsibilities of individuals in different positions. For instance, do the ethics of a CEO differ from the ethics of employees, citizens, politicians, etc.?

It is important to reiterate that these interviews were conducted as the first phase of a Design Research Project intended to guide the curricular design for Critical Mathematics Consciousness. As such, the individual nature of the interactions does not speak to the learning that results from student collaboration and class discussion. They also did not teach new mathematical concepts. However, that was not the intention of the phase. Rather, in line with its purpose, the task analysis provided important insights for Curriculum Design which will be discussed in the next section.

### 5. Discussion and Design Implications

The purpose of the first phase of this design research project was to identify contextual factors of tasks that elicit Critical Mathematics Consciousness in privileged students. The students involved in the interviews were either White, socioeconomically, or educationally privileged in that they all attended either a private or charter school of choice. The promotion of equity is a collective and collaborative responsibility. It involves actively working towards fair and honest practices and dismantling discrimination at the micro and macro levels. K-12 classrooms serve as a laboratory for exploring such issues in contexts that are familiar and relevant to students' lives. Curricula in K-12 mathematics classrooms must

draw on such contexts both because of the social implications of developing ethical mathematicians/citizens and to support "productive engagement" with mathematics [57–60]. Task design is essential to these outcomes.

The two tasks discussed in this paper, *Corona Crisis* and *Great Groceries* evoked students' sociopolitical, ethical, and communicative awareness (CMC) and thus provided considerable insight for our next steps in the project and for research more broadly. Such insights include that the initial tasks must be accessible to all students, that they must first build their communicative MA and ethical MA before considering sociopolitical issues that may be outside of the scope of their experiences, and that the tasks should involve ethical decision-making processes. These will be discussed in the sections below, followed by our future design considerations.

### 5.1. Where to Start: Accessibility

We hypothesize that tasks like *Corona Crisis* could serve as a viable starting point for instruction, due to its accessibility to all students. In our analysis, we found that our participants could "see themselves" in the COVID-19 graphs (i.e., in the data), enabling them to become mathematically and critically engaged. Since students were experiencing the spread of the virus at the time of the interview, their mathematical interpretation of the task had consequences for their personal safety and the safety of those for whom they care. As a result, the students more readily recognized manipulation by the authors of the graphs. While we are certainly not positing that all ethical mathematics learning should begin with a task about the coronavirus pandemic, we do want to draw attention to the attributes of the *Corona Crisis* task that we feel make it an ideal starting point for privileged students: (1) the global nature of the task, (2) familiarity with the mathematical representations used, (3) the informal inference required to access it, and (4) the focus on communicative MA.

First, the global nature of the context enables students of all races, genders, sexualities, socioeconomic statuses, etc., to understand its relevance to their lives. Although not discussed in this paper, we analyzed a task related to race and police induced deaths as the first task of the interviews. While issues of race and police shooting were (and are currently) a controversial topic in the U.S., most of the students interviewed could not draw on their experiences to access the task (being White and/or socioeconomically privileged). In contrast, the *Corona Crisis* task was cognitively and emotionally accessible to all students.

Second, the mathematics conceptualizations that students were asked to perform were based on visual representations that are common to everyday citizens. This enabled students to see their mathematical reasoning as a utility for navigating a real-world phenomenon. Third, students were not expected to learn or perform mathematical procedures. Rather, in line with exploratory data analysis (EDA) and informal statistical inference (ISI), students began the task by conjecturing and asking questions about real world data [58]. This enabled them to make connections between the data and the broader context where the data was collected, thus connecting mathematics to their lives. This aspect also encouraged appropriate norms for an inquiry-driven learning environment dependent upon discussion, analysis, and reflection.

Finally, the *Corona Crisis* task was intended to promote communicative mathematical awareness. Rather than beginning with a task that is sociopolitical in nature, this task was concerned with how mathematics can be used to promote certain agendas. As a result, students began to ponder the prevalence of bias in their sources of information (social media, television and web-based news sources, parents, peers, etc.), a consideration that we feel might open their minds to more controversial topics in succeeding tasks. As an example, proportional reasoning is a notoriously difficult mathematical concept [59] whereas interpreting a line graph is more accessible for students. Once students are primed to consider that mathematics has the power to (mis)educate, a subsequent task could extend the COVID-19 data to include the number of infections broken down by ethnicity to engage students in a conversation about systemic racism. After students see the disparity of COVID-19 infections within communities of color, they might begin to ask questions about

other oppressed communities (e.g., people in poverty) or about other kinds of inequities perpetuated on communities of color (e.g., police brutality, credit denial, etc.).

*5.2. Ethical Decision Making*

In a White Paper arguing for the integration of ethics into mathematics education, Chiodo et al. [60] warn against the consequences of failing to provide ethical training to mathematicians. They highlight that, unlike students in other technical subjects, most mathematics students are not given the opportunity to contemplate the possible legal, ethical, and social implications of their work, despite the reality that it affects the lives of others. Typically, undergraduates who study mathematics in the 21st century do not end up working in academia, but instead go on to occupy positions that play a fundamental role in the economy and in society, using their mathematical skills in a manner that has ethical repercussions. The absence of ethical training in mathematics education often results in the adoption of early employers' and colleagues' ethical dispositions (or lack thereof). To make matters worse, the frantic culture of the modern workplace does not allow its employees time to consider issues outside of their immediate frame of reference. Thus, it is important that education in abstract technical fields such as mathematics includes exposure to the potentially harmful consequences of students' mathematical work as well as the development of the skills and instinct to identify ethical issues in the future [60].

The ethical preparation of career mathematicians and critical citizens requires that students be exposed to ethical decision-making processes at an early age. As such, mathematical tasks should involve real world contexts and require students to justify the most ethical decision using mathematics (in their opinion), as opposed to the most mathematically sophisticated solution. *Great Groceries* is one such task. While representing another task accessible to all students, *Great Groceries* was premised on ethical decision-making in an economic context. It required that students use their mathematical and contextual conceptions to make a decision which would have societal consequences as opposed to analyzing a phenomenon which had already occurred (as in many TMSJ tasks). Importantly, the task was open ended and did not have a correct answer. Instead, the expected response is a subjective one based on students' experiences and mathematical understanding. That being said, the question raises important questions about what it means to be ethical in the business world and how expectations of ethical behaviors may change according to who is in charge. Questions such as these, where students are required to ponder the effects of their mathematical solutions in society, are essential for promoting ethical mathematicians.

*5.3. Progression from Ethics to Social Justice*

As stated in relevant literature, the development of Critical Mathematics Consciousness in students of advantaged groups differs from the development of CMC in students from minoritized groups due to their differences in experience. Sociopolitical consciousness is not a quality that can be given to people via any means. Rather, they must develop it themselves. With this in mind, we believe it may be beneficial for students of privilege to experience tasks that progress through and build communicative MA, followed by ethical MA, and finally, sociopolitical MA. It is important that students of privilege are able to draw on their own experiences throughout this process, and especially at the beginning of such a learning trajectory. In our experience, it was useful to provide a task that was accessible to all students (*Corona Crisis*) and which first focuses on the development of the communicative MA. By encouraging them to recognize the potential lack of fairness, dishonesty, and/or intentional miscommunication in their own lives, we felt that it may have opened their minds to the possibility of such phenomena in others' lives. The next step in the progression is to build their ethical awareness. In accordance with literature on ethics, issues of social justice are inherently ethical and depend on empathy for others. Thus, before engaging privileged students in tasks intended to analyze social injustices, it may be necessary to have them ponder and make ethical decisions that post consequences for the self, others, culture/society, or various ecologies (e.g., *Great Groceries*). By encourag-

ing students of relative privilege to consider the implications of decisions in the context of real-world scenarios, students may be more open and able to recognize instances of oppression, enabling them meaningfully to engage with social justice tasks and to develop sociopolitical awareness.

### 5.4. Moving Forward

As stated previously, the data analysis in this paper represents the first phase of a Design Research project intended to determine task contexts that might elicit students' CMC, including their ethical awareness. We believe that the tasks highlighted in the analysis section provided considerable insight into the next steps for our design. Specifically, the initial tasks must be accessible to all students, they must first build their communicative MA and ethical MA before considering sociopolitical issues that may be outside of the scope of their experiences, and should involve ethical decision-making processes. In addition, and to promote student engagement in these tasks, they must be mathematically, technologically, and contextually provocative [61]. In the context of effective statistics learning, Madden [61] defines "provocative tasks" as tasks that create a "cognitive hook" for students. They include statistically provocative tasks (the statistics are interesting), technologically provocative tasks (the insights they gain from the power of technology are interesting), and contextually provocative tasks (the context is intriguing, familiar, and something they wish to understand) [58,61]. While *Corona Crisis* and *Great Groceries* provide insight into what may be considered contextually and mathematically provocative for privileged students, they do involve technology in a meaningful way. As such, our next steps for task design will be to create and/or adapt the task of the learning segments that engage students with technology. We also hope to take an interdisciplinary approach and include authentic computer and data science concepts. Finally, we recognize the importance of peer collaboration and an interactive learning environment. As such, we aim to develop inquiry-driven, discourse-based, and collaborative lessons that will be used to facilitate ethical mathematics learning using the derived tasks.

### 6. Conclusions

In this article, we argued that didactics of mathematics in the United States has undergone a radical shift towards researching and teaching mathematics with an equity focus. Many researchers have called for equitable access to mathematics and recommend adjusting curricula to connect better to students' lived experiences outside of school. To that end, many didacticians, including curriculum designers, teachers, and researchers have shifted their agendas to reflect a social justice pedagogy in which students "read their world" mathematically and suggest ways "to re-write" it in a fairer way. We presented a brief description of the findings from research on critical mathematics pedagogies, noting significant strengths and areas for further research. Since there is little empirical research on students' ethical reasoning in mathematics, we introduced the Ethical Reasoning in Mathematics Framework and presented the findings from the initial phases of a Design Research project to illustrate how the Framework can be used to document students' ethical reasoning. These findings led us to make some recommendations for mathematics curriculum designers and teachers who want to infuse ethical reasoning into their instruction. With the rise of big data as a mathematics discipline, there is great potential to perpetuate inequities among certain groups in society whether knowingly or not. Therefore, increased attention to ethical reasoning in mathematical decision-making is critical and deserves a more robust empirical research knowledge base to inform the didactics of mathematics.

**Author Contributions:** Conceptualization, J.R., M.S. and D.P.; methodology, J.R., M.S. and D.P.; formal analysis, J.R., M.S. and D.P.; investigation, J.R. and M.S.; resources, J.R., M.S. and D.P.; data curation, J.R. and M.S.; writing—original draft preparation, J.R. and M.S.; writing—review and editing, J.R., M.S. and D.P.; visualization, J.R.; supervision, M.S. and D.P.; project administration, M.S.; funding acquisition, D.P. All authors have read and agreed to the published version of the manuscript.

**Funding:** This research received no external funding.

**Institutional Review Board Statement:** The study was conducted according to the guidelines of the Declaration of Helsinki, and approved by the Institutional Review Board (or Ethics Committee) of the University of North Carolina at Charlotte (IRB Proposal #19-0594 approved 2/17/2020).

**Informed Consent Statement:** Informed consent was obtained from all subjects involved in the study.

**Data Availability Statement:** The data presented in this study are not publicly available in any form due to the subjects being minors and signing assent forms acknowledging that full videos will be seen by only researchers listed on the approved IRB application. De-identified data in written format can be requested from the corresponding author.

**Acknowledgments:** We acknowledge Premkumar Pugalenthi, Christine Robinson, Luke Reinke, for their intellectual support on this project; the University of North Carolina at Charlotte Center for STEM Education for internal funding and logistical support; and the students, teachers, and administrators at the sampled schools for accommodating our desire to conduct this important research.

**Conflicts of Interest:** The authors declare no conflict of interest.

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
