# Peer review of "Ethical Reasoning in Mathematics: New Directions for Didactics in U.S. Mathematics Education"

_mathematics, doi:10.3390/math9080799_

Round 1

Reviewer 1 Report

In the review of literature there was no mention of the work of Ole Skovsmose, the leading European researcher on Critical Mathematics Education. This should be rectified

Author Response

Ole Skovsmose's name (as opposed to just a citation) was provided as a seminal researcher in critical mathematics (page 4 of manuscript). We agree that leaving his contribution out was a major error on our part. Thank you for this.

Reviewer 2 Report

My primary (multi-faceted) concern is in following the connections between the various analytical tools introduced by the authors and the analyses and discussions in the paper.

Firstly, my understanding of the main goal for this paper is to introduce and apply their ERiM framework, with its 12 ethical principles and corresponding ethical considerations (in the form of questions). However, after a lengthy, and frankly excellent, review of the literature in CME/CMP, including identifying the gaps into which their study could fit, the authors ‘shortchange’ the reader by providing very sketchy details as to how the framework came about. It is not clear how/why these 12 ethical principles became the structure for the framework (sources were identified in general, in the third column of the framework table, but no specifics provided). Additionally, the ethical considerations column draws on language of data, system, organization (presumably language to align with their source) but the authors do not explain this language nor do they make an effort to align the language within the table (for example, each consideration/question for each ethical principle should draw on similar language so that a reader can see the distinguishing features of each principle across all 12). Truthfully, the language used in this table is much more based in quantitative forms of research than the qualitative goals these authors appear to have.

Secondly, while I can appreciate the authors’ definition of CMC, including its corresponding 3 types of mathematical awareness (MA), the connections between this and the framework is not at all clear. Thus, when I read the analysis of the first task (Corona Crisis), I was surprised to find reference to three themes emerging from the data, which were not related in any explicit way to the framework (and ethical principles) although one of the themes was the third type of MA. This approach to analysis perplexed me. I read through to the end of this section (4.1) and still no mention of the ethical principles and the framework. Then, a dramatically different approach was taken for the analysis of the second task (4.2 Great Groceries) which actually had a section immediately following (4.2.2) that was titled Ethical Considerations (I think this title should have been Ethical Principles because that is what the authors go on to identify in that section). My concern then, in a nutshell, is that the paper lacks coherence between the sections of the paper; the tools identified are not applied in a consistent manner using consistent language (perhaps the authors divided up the paper and then did not address alignment and coherency between the various parts appropriately)

The paper was so well-conceptualized and written in the first 12 pages that it was disappointing to then become so confused once the methodological/analysis tools were introduced and applied. My recommendation would be to focus on the framework, both in terms of how it was constructed and then what it offers when applied to the interview data. Focusing on only two of the five tasks for analysis seems fine but both tasks should be analyzed through the same lens (of the framework).

Author Response

  • Line 473: added graphic to clarify how the CMC and ERiM frameworks fit together. 
  • 486-494: Written clarification of how the frameworks fit together. 
  • 532-541: added clarification to ERiM Framework description.

We hope that the graphic in Figure 1 will help explain the relationship among the different mathematical awarenesses involved in CMC and why some of the data warranted foregrounding the ethics framework while backgrounding the sociopolitical and communicative and other data warranted different foregrounding/backgrounding. 

Reviewer 3 Report

I believe that this is a very interesting work for the educational community of Mathematics and that it focuses on an aspect that is not attended to in teaching, since it has been interpreted that mathematics does not enter into ethical aspects.

Here are some comments in order to improve the manuscript.

The reading of it is sometimes not clear.

The literature review sometimes offers some opinion (line 64, line 103) or at least is presented in an unclear way. No clear research question presented. In the work, a framework is offered to analyze ethical reasoning in mathematics, an implementation with some ethical principles is carried out and it is analyzed. But the connection between the entire process developed is not well explained and it is not clear if the objective of this work is to present the framework, test the framework or analyze student responses from ethics.

I believe that there are some gaps in the discourse that must be better connected. It jumps from the analysis of training for social justice to a framework for ethics in mathematics education and from here to a framework for the analysis of ethical reasoning. The data that are analyzed does not specify the instrument, the methodology followed, how the results were agreed between the researchers. It does not offer a clear research question and conclusions that answer or are close to the answer of the proposed research, ...

I believe that the manuscript should be reviewed in depth.

Here are some specific improvements or doubts that arose during your reading:

Line 69. Uppercase letters in some moments and line 74 for the same terms in lowercase letter.

Line 99. JRME does not indicate what it refers to

Line 113. It is not appropriate to write a web address in the body of the text. Use a webgraphy for it

Line 117. Select the term didactics or pedagogy, but do not modify the terms during the speech. Why not use Math Education?

Line 161. I consider that political knowledge is not a concept coined by Gutierrez, but rather it is knowledge about politics. Therefore, it should not be considered in Spanish, but rather its translation into English.

Line 170. When mention is made of Freire, his reference must be indicated, since it is not clear from where this information is obtained.

Line 178. The reference for this statement is not offered, which is very general and is not verified in this work.

Line 184. When you talk about oppressor today, what are you referring to? Something so general has not been specified: political, racial, cultural, experiential, ...

Line 188. Why are CMP and CR said to have been impactful? Justify this sentence

Line 196. Structural / Curriculum… Why that bar? Change it to "or". It is not well clarified what is intended when it comes to structural requirements. In addition, in the discourse that is developed in this section it is discussed how evaluation affects, no reasons are given about the curriculum. So it does not correspond to what is concluded.

Line 219. In addition… I don't understand what is meant by this sentence and how it relates to the section that is being developed.

Line 236. See comment line 161

Line 247. I would not put the example in parentheses

Line 257. Error with quotes

Line 291. See comment line 99

Line 343. Omission of quotation marks at the beginning

Line 348. Omission of quotation marks at the beginning.

Line 362. Quotes are left over.

Line 370. It is commented that there are differences between the teaching of mathematics for social justice and that there are few studies that explicitly focus on the others as ethical imperatives for mathematics. I do not understand what this last sentence refers to (line 371)

Line 386. If you name an author, you must indicate their bibliographic reference to develop the correct connection.

Line 400. David Stinson reference missing

Line 477. The framework of ethical reasoning in mathematics is offered in table 1. But this framework is not connected with ERiM, nor is it explicit why these ethical principles have been considered. How are the considerations obtained? Some of the domains are linked to associations such as the ASA, but I do not understand that privacy, for example, is linked to this association and does not have a specific research work that promotes it as a reference. I believe that it is necessary to better explain how the proposal of this framework is made, as it is essential in the analysis that is then carried out.

The considerations are posed as questions. How have these questions been constructed and why do these questions collect all the necessary information about the principle being analyzed? Are these questions valid for the educational field?

To what extent is this framework typical of mathematics? I do not observe that this framework has characteristics of the teaching of mathematics, and it is not explained why they consider it that way. I see it necessary to dedicate more space to explain the origin and construction of this instrument.

Line 493. I don't understand what the phrase… distinctions between fairness and discrimination… is the only principle that needs explanation? Why?

Line 498. It is not explained how the sample of participants is selected. Nor are the data on race, origin, ... which are understood to be of interest for this investigation. The students are not identified and each of them is discussed later. The reader does not understand the relevance of one student to another or in which category they are. This explanation appears on line 943 to line 953. I think you should move to this place.

The information gathering instrument is not explained. You need to make it explicit before. The tasks are described on line 954 to line 961. I also consider that it should be in this section, not at the end of the manuscript.

Line 507. Error: 13 students, then 15 are spoken

Line 520. Collection of pilot data. But it is the data that is going to be used in the work, it should not be indicated that it is pilot.

Line 528. See comment line 113

Figure 1. Inside the image: Corona Crisis Part 1 and Corona Crisis Part 2 are indicated, but the figure does not identify what each part corresponds to.

Line 572. The ERiM study is carried out, describing the activity and commenting on the students' responses. But it has not been specified previously which analysis instrument is being used. The ERiM are general categories, but it has not been related to ethical aspects

Line 627. How many students did not notice a difference between the graphs?

Line 704. Improve the quality of figure 2.

Line 686. Great Groceries. This section is not analyzed from the ERiM perspective. Fairness, discrimination, employment and loyalty are studied. Why is the analysis done from this perspective? It has not been indicated at any time that only part of table 1 should be used. Why these and not others?

Is decision making about companies, salaries, employability, ... considered close to students? If the subject is not close, the answers about the risk will be due to other factors that will not be easy to link to those of their social environment. That is, to what extent are the answers given by the students due to the aspects that are intended to be analyzed in this work?

Line 784. A statement from a student is highlighted. But it is not clear how important this is and, above all, what information this answer is giving us.

Line 872. I think it should go to the next point, within 4.2.3

Line 942 and ss. The characteristics of the participants are explained and the tasks are explained. I do not consider it to be a discussion based on the data obtained. Nothing is said about the results of the students, the information they give in their decision-making, nor is it related to the analysis instrument used. I would need a table that collects the information, a diagram that explains the process followed, ...

Line 1076. Provocative tasks. I think it is a very interesting feature to take into account from the beginning of the proposal.

I hope that my comments serve to improve your work.

Author Response

Thank you to this reviewer in particular for such a thorough review. 

Reviewer 4 Report

Comments to “Ethical Reasoning in Mathematics [ERiM]: New Directions for Didactics in U.S. Mathematics Education ”:

- What is the objective of this manuscript? It should appear in the Intro. Even describe explicit objectives and actions
- An excessive use is made of the personal form (we) in the writing. The scientific article must overcome this barrier and be written in an impersonal way.
- "The term didactics can refer to either the practice of teaching mathematics or the research of mathematics education depending on the cultural and educational background of the writer." What literature have you used to provide this definition?
- Section 2 is well planned and written, although due to the density of documentation provided, it is convenient to make a graph that provides clarity in the reviewed literature and what its function is in the manuscript.
- Can you indicate in which other jobs this method has been applied successfully?
- A global discussion is necessary in section 5
- Why do you use both work and article in all the text? Decant for one of them
- One of the main drawbacks observed is the excessive literature provided that distorts the purpose of the manuscript

Author Response

In lines 43-45 we include a sentence that outlines the purpose of the paper more explicitly.

We attempted to remove the more personal pronouns but left those that seemed necessary to humanize the research. 

Reviewer: wrote: The term didactics can refer to either the practice of teaching mathematics or the research of mathematics education depending on the cultural and educational background of the writer." What literature have you used to provide this definition? We were not trying to provide a definition exactly; we were only trying to clarify that the term is used in both ways according to the research we have read. We do not distinguish between the two uses of the word in our paper, and just wanted to state that upfront. Since we already have over 60 references, we worried about adding more references on didactics. Either we can take that sentence out, or leave it with our short revision: Based on our reading of didactics articles by authors from a variety of countries, [line 59]

Regarding creating a picture to convey the details in section 2, we made one to summarize the major work on ethics and critical theories (see figure 1).

This method is being used for the first time in this paper. 

Regarding a more global discussion in section 5, we were not sure what was meant by this. Our discussion section began with general discussion about the purpose of the study and then moved to implications of the results. We are willing to rework this if we have more guidance.

We have conducted a search of the manuscript to identify all places where the word "work" was used and changed them either to "research", "article" or left them the same if it was warranted.

We cut down on some of the text but not a lot. We received mixed reviews on the literature review. One reviewer said it was exceptional and this review said it was too much. So, we tried to cut text that may have been excessive.

Round 2

Reviewer 4 Report

Comments on the manuscript entitled "Ethical Reasoning in Mathematics [ERiM]: New Directions for Didactics in U.S. Mathematics Education":

- It is not necessary to include the acronym in the title. This is what the text is for.
- Why do you put the abbreviation of United States in the abstract, but on line 24 (when it first appears) you don't indicate this abbreviation in parentheses? Correct it
- Line 29: "now known as the Research Conference". Since when? Where is this indicated?
- I keep seeing that 12 pages of Intro + Literature Review is too much. You need to condense it
- The rest of the paper is very well worked. I congratulate you for it

Author Response

- It is not necessary to include the acronym in the title. This is what the text is for. : Acronym in title removed. 
- Why do you put the abbreviation of United States in the abstract, but on line 24 (when it first appears) you don't indicate this abbreviation in parentheses? Correct it: (U.S.) added in parenthesis on line 24. 
- Line 29: "now known as the Research Conference". Since when? Where is this indicated?: This line removed completely in consolidation of Intro and Lit Review. 
- I keep seeing that 12 pages of Intro + Literature Review is too much. You need to condense it: Condensed to 8 pages
- The rest of the paper is very well worked. I congratulate you for it: Thank you!